# “You Feel a Sense of Accomplishment”: Outdoor Adventure Experiences of Youths with Visual Impairments during a One-Week Sports Camp

**DOI:** 10.3390/ijerph20085584

**Published:** 2023-04-20

**Authors:** Lauren J. Lieberman, Katie Ericson, Melanie Perreault, Pamela Beach, Kelsey Williams

**Affiliations:** 1Department of Kinesiology, Sport Studies & Physical Education (KSSPE), State University of New York Brockport, 350 New Campus Drive, Brockport, New York, NY 14420, USApbeach@brockport.edu (P.B.); kwill24@brockport.edu (K.W.); 2Department of Special Education and Communication Disorders, University of Nebraska at Lincoln, 301 Barkley Memorial Center, Lincoln, NE 68583-0738, USA; katherine.e.ericson@gmail.com

**Keywords:** recreation, blind, modification, adaptations, low vision

## Abstract

There are many cognitive, physical, and social–emotional benefits for youths from participating in outdoor adventure activities. However, youths with visual impairments are not given the same opportunities to participate in outdoor adventure activities as their peers without disabilities. The purpose of this study was to examine the outdoor adventure experiences of youths with visual impairments participating in a week-long sports camp. Thirty-seven youths with visual impairment (ages 9–19 years) attending a one-week sports camp participated in this study. Participants engaged in a variety of outdoor adventure activities throughout the week of camp (e.g., sailing, hiking, rock climbing, biking, kayaking). Participants provided written accounts about their outdoor adventure experiences and were observed throughout the week during each activity to examine instructional strategies and task modifications. Additionally, 10 randomly chosen athletes, their one-on-one coaches, and five outdoor recreation specialists participated in focus group interviews. The data analysis revealed three major themes: (1) Benefits, (2) Support, and (3) Barriers. The subthemes of benefits were enjoyment, independence, and relationships; the subthemes of support were instructional strategies and task modifications; and subthemes for barriers were fear and anxiety, exclusion and low expectations, and lack of equipment. These findings support the inclusion of youths with visual impairments in all outdoor adventure programming with appropriate instruction and modification.

## 1. Introduction

Outdoor adventure experiences benefit individuals’ cognitive, physical, and social–emotional well-being. In particular, outdoor adventure experiences can improve not only an individual’s physical health, but also self-concept, self-determination, social skills, and mental health [1,2,3]. Wood and colleagues [4] indicated that outdoor recreation can decrease symptoms of anxiety, depression, and stress. Outdoor adventure experiences do not have to involve a high degree of challenge or expensive equipment; however, many programs encourage individuals to set and achieve challenging goals [2,3]. 

The benefits of outdoor adventure programs extend to individuals with disabilities, particularly individuals with visual impairments [2,3]. Successful participation in outdoor adventure experiences can promote independence and motivation, increase environmental knowledge and perceived competence, and lead to feelings of personal empowerment [2,5,6,7,8]. In a recent study, individuals with visual impairments reported feelings of personal freedom when engaging in physical activity in nature [7]. 

Many benefits of outdoor adventure experiences align with components of the Expanded Core Curriculum (ECC). The ECC includes nine skill areas that most youths acquire through incidental learning but must be explicitly taught to students with visual impairments [9]. These nine skill areas encompass assistive technology (AT); career education; compensatory skills; independent living; orientation and mobility (O&M); recreation and leisure; self-determination; sensory efficiency skills; and social interaction. A study conducted by Lieberman et al. [10] indicated that engagement in physical activity promoted all nine components of the ECC skill development. This research is compelling, although there has been limited research in the area of youths with visual impairments and outdoor adventure to date. 

Despite these potential benefits, individuals with visual impairments participate in outdoor adventure activities at lower rates than their peers without disabilities [2]. Lower participation is in part due to fewer opportunities for youths with visual impairments to participate in physical activity and possible exclusion from more traditional physical activity settings [2]. Additional barriers may include social–emotional, financial, or structural constraints [2,3]. For example, young athletes with visual impairments may not be able to afford needed equipment, such as fishing reels, hiking boots, or a stand-up paddleboard. Even when youths with visual impairments surmount these barriers, they may still confront a fear of the new and unknown [11]. 

Although many youths and young adults with visual impairments experience barriers to outdoor adventure activities, participation can be beneficial to students and facilitate instruction in the ECC. Considering the potential benefits of outdoor adventure activities for youths with visual impairments, the purpose of this study was to explore camp athletes’ experiences with such activities at a sports camp for youths with visual impairments using a qualitative approach. Within this paper, we consider outdoor adventure experiences such as hiking, biking, stand-up paddleboarding, kayaking, rock climbing and sailing. In addition, we seek to determine how the Camp experiences compared to their previous experiences in outdoor adventure.

The research questions were:What are the experiences of youths with visual impairments related to outdoor adventure during a one-week sports camp?What accommodations or supports help youths with visual impairments access outdoor adventure activities?How did these experiences compare or inform their previous and future experiences in outdoor adventure programming?

## 2. Materials and Methods 

### 2.1. Participants

Participants were recruited from 37 campers attending a one-week overnight educational sports camp for youths with visual impairments. Participants included 10 athletes with visual impairment aged 9–19 years, with an average age of 13.2. There were 4 females and 6 males. Their visual impairment ranged from B1–B4 as recognized by the USABA sport classifications: “B1 has no light perception in either eye up to light perception, and an inability to recognize the shape of a hand at any distance or in any direction, B2 has the ability to recognize the shape of a hand up to visual acuity of 20/600 and/or a visual field of less than 5 degrees in the best eye with the best practical eye correction, B3 has visual acuity above 20/600 and up to visual acuity of 20/200 and/or a visual field of less than 20 degrees and more than 5 degrees in the best eye with the best practical eye correction, and B4 has visual acuity above 20/200 and up to visual acuity of 20/70 and a visual field larger than 20 degrees in the best eye with the best practical eye correction”. See Table 1 for specific descriptive information for each athlete in the focus groups. Institutional Review Board permission was granted prior to the study. Each participant in the focus group and their parent/guardian provided consent for participation prior to data collection.

### 2.2. Procedure

Participants experienced a variety of outdoor adventure activities throughout the week at camp including sailing, hiking, rock climbing, kayaking, stand-up paddleboarding, biking, and high ropes elements. During the week, three main sources of data were collected to understand how participants experienced the various outdoor adventure activities. These sources of data included written accounts, observations, and focus groups.

#### 2.2.1. Written Accounts

Throughout camp, posters were available in the main common area of the camp dorms for participants to record their thoughts about the different activities as well as adaptations and modifications they experienced. Each poster had the name of one of the outdoor adventure activities and blank post-it notes as well as cards to add braille that athletes could post. Participants recorded their responses using black markers on the post-it notes and stuck them on the corresponding activity. In addition to the posters, a clipboard with paper and pens was set up at each outdoor activity, and researchers encouraged participants to write down their feelings, adaptations used, as well as instructional strategies that worked or did not work. All participants were invited to share their feelings about the outdoor adventure as well as the modifications and instructional strategies that worked or did not. Most of the campers shared their feelings about at least one activity according to anecdotal feedback from their coaches. The true number is not known, as this step was anonymous.

#### 2.2.2. Observations 

Four of the researchers collected field notes during all of adventure activities and recorded interactions with the coaches and their peers, instructional strategies, feedback, and adaptations used for each activity. These four researchers documented adaptations (equipment, instructional strategies, positioning, etc.), comments heard related to involvement in the activity, and any socialization aspects. One to two researchers attended each activity on a daily basis throughout the week.

#### 2.2.3. Focus Groups

Ten of the athletes and their one-on-one coaches were randomly chosen to participate in focus groups (see Table 1 for descriptive information about each athlete). Each athlete agreed to participate in the study. Five of the outdoor recreation specialists were also invited to participate in the focus groups. Participants were divided into four focus groups. Two of the focus groups were made up of five athletes each. One focus group included seven of the one-on-one coaches and another consisted of the five recreation specialists. Please see Appendix A for the focus group scripts for the athletes and coaches. During the last day of camp, the four focus groups convened to answer questions and discuss their experiences. The focus group questions were developed by the researchers and reviewed for face and content validity by five specialists in the field. The specialists consisted of one individual with expertise in visual impairment and qualitative research; two individuals with expertise in motor development and visual impairment; and two individuals with expertise in outdoor adventure and visual impairment. The questions for the athletes focused on the participants’ experiences in outdoor adventure, what worked and what needed to be improved, and how they would access outdoor adventure when they went home. The questions for the coaches and specialists focused on the experiences of the athletes during outdoor adventure, modifications used, instructional strategies, enjoyment, and socialization.

### 2.3. Data Analysis

Each focus group recording was transcribed verbatim. Researchers also transcribed notes from the posters and clipboards, combining these in one document, with specific data grouped by activity. The observations were also transcribed and grouped with each activity. The focus group transcripts and all of the data collected were analyzed by two researchers. First, the two researchers independently coded the data for themes, subthemes, and accompanying quotes to fully understand the entirety of the transcript data as well as clipboards, observation, and poster data [12]. The researchers used Braun and Clark’s [12] recommendations related to completing thematic analyses to ensure their analysis was performed in a theoretical and systematic manner. Upon completing their initial coding, the two researchers met to discuss variations of codes, assess common codes, and review emerging themes. During these discussions, the themes and subthemes were developed and continuous analysis of the data through repeated examination occurred [13].

Lastly, two “critical friends” who were experts in qualitative research and outdoor adventure read each of the transcripts as well as the themes and subthemes to ensure all themes were reflective of the data collected and in alignment of what the participants shared in the focus groups. The fact that the two researchers conducting the study were outdoor enthusiasts and knew some of the participants, meant that the unbiased perspective of two additional researchers outside of the program who were familiar with the field was paramount [14]. These “critical friends” held the researchers accountable and helped to ensure an unbiased lens.

Once the four researchers agreed upon the findings, the major themes and subthemes were reduced based on similarity of meaning and content. The themes and subthemes along with the quotes presented below as findings were agreed upon by all researchers.

#### 2.3.1. Trustworthiness

In this qualitative analysis, researchers used several strategies to ensure trustworthiness. The researchers conducted four focus group interviews as well as observations and written accounts (posters and clipboards at each activity) on activity modifications and how the participants felt during each outdoor adventure activity. These three methods of data collection support triangulation of the data [12,13]

In addition, at the beginning of each focus group interview, the researchers exposed their positionality. This explanation of their lens gave the participants insight related to their professional and personal perspective on the topic of outdoor adventure [14]. Moreover, two “critical friends” were utilized when analyzing the themes and sub-themes to acknowledge and reduce researcher bias [15,16]. Due to these systematic measures taken to ensure accuracy, the researchers believe the data gathered represents a true reflection of what participants experienced during the outdoor adventure activities at camp.

#### 2.3.2. Findings and Discussion

This study revealed three major themes of outdoor adventure experience for youths with visual impairments during a one-week camp: (1) Benefits, (2) Support, and (3) Barriers. Benefits included three subthemes: enjoyment, independence, and relationships. The subthemes emerging from support included instructional strategies and task modifications. Three subthemes emerged from barriers: fear and anxiety, exclusion and low expectations, and lack of equipment. 

### 2.4. Benefits

Most participants, according to the focus group interviews, had previous experience with outdoor adventure activities and actively participated in these activities during camp. Similar to previous research, many benefits were found to be derived from outdoor activities [7,10]. Both athletes and coaches discussed many benefits of outdoor adventure activities, which resulted in three subthemes: (1) enjoyment, (2) independence, and (3) relationships.

#### 2.4.1. “It Just Feels Nice”: Enjoyment

Most athletes described their enjoyment of outdoor adventure activities, which is in line with previous research wherein increased enjoyment, satisfaction, and well-being was derived from participating in outdoor activities in comparison to indoor activities [1,17]. Even walking outdoors can enhance moods and increase motivation to continue walking in comparison to walking indoors [18]. Dennis shared, “It just feels nice. Going outside, such as in the forests, rivers, lakes, is just fun”. Elena agreed, saying, “I feel good and can say I did it. You feel a sense of accomplishment”. Their coaches and specialists echoed these sentiments. Elena’s coach stated, “I think mostly excitement would be the word for Elena”, and Alexander’s coach shared a similar statement: “I would say excitement, and anything outdoors, it was such a change on what they are used to … when they went outside it was a complete game changer”. 

#### 2.4.2. “Tandem Gives Kids Freedom”: Independence

Other participants noted the sense of freedom and independence they felt. One of the tandem biking specialists shared that “tandem gives the kids freedom”, and she observed that there seemed to be more independent bikers this year than in the past. Alexander stated that nighttime activities, which were “free choice”, allowed athletes to be successful because “you could go out and do what you wanted to do with you and your coach and friends”. When asked what his favorite activity was, Alexander elaborated, “Fishing because you are by yourself, it takes a while, it is calming, relaxing, and quiet”. Similarly, Dennis reported stand-up paddle board was his favorite outdoor activity because he liked “moving fast and piloting my own ‘ship’”. The researchers also observed that many of the participants became independent in rowing, rock climbing, and in biking.

Independence (one facet of self-determination) is one of the components of the ECC [9], and Lieberman and colleagues [10] indicated that engagement in physical activity promoted all nine components of the ECC including O&M, recreation and leisure, self-determination, and social skills. Increased self-determination may lead to higher intrinsic motivation to accomplish more both inside and outside of the classroom, improving their academic results and health [19]. The independence derived from participating in outdoor activities can boost a child’s self-confidence. The freedom that comes from independence can boost their self-confidence to accomplish goals that they otherwise may have viewed as unobtainable [2].

#### 2.4.3. “OMG, I Want a Friend like You”: Relationships

Similar to the results from Lieberman and colleagues’ [2] research on outdoor education with youths with visual impairments, participants in the present study described psychosocial benefits of being outdoors, such as the opportunity to be with friends and family. Robert stated, “I like hiking because I can grab a couple friends and just talk while we walk around”. Elena liked fishing for similar reasons, saying, “There was a group of people, so we all chatted, and I love that”. Their coaches also noted how important these social experiences were. Elena’s coach observed that Elena “always wanted to match with her friends, to see what activity they were doing to do it with them”. Similarly, Robert’s coach reported, “I think he [Robert] was hype to be a part of the tournament and to play with his friends that were in the same group”. On the post-it notes on the posters in the dorms athletes said they liked sailing and stand-up paddle board because they could be “active with friends”.

All athletes must have a visual impairment to participate in camp, and this shared identity creates a unique sense of community not often experienced in other settings. For example, Robert’s coach observed, “… the other thing is they are most likely the only kid with a visual impairment in their school and they come here, and they do not stick out. Accommodations no one thinks twice about it, and it so cool when they walk around and talk about the similarities/differences of their vision.”

Andrei’s coach agreed, reporting, “Andrei found it so cool that one of the Paralympians had his exact same disability”. Dennis’ coach shared, “Dennis had his braille reader and he let Alexander play with it. Alexander stated, ‘omg, I want a friend like you’. It was great to see the joy in his face with someone else who had complete blindness”. The observations also revealed that the participants felt fine with physical and verbal assistance because many of their peers “needed the same instructional strategies”.

### 2.5. Support

All athletes required some type of support to successfully participate in outdoor adventure activities. The theme of support highlights the modifications and instructional strategies that facilitate youths with visual impairments to be able to engage in outdoor recreation experiences [2]. Engaging in outdoor recreation activities with modifications is directly linked to ECC components [8,9]. Youths with visual impairments can improve their mobility and independent living skills through outdoor activities while also receiving many other ECC benefits discussed above including social skills, recreation, and independence. 

When asked, most athletes remarked that the supports they most appreciated were emotional; they thrived on their coaches’ words of encouragement. As Elena noted, “Motivation helped me as well; everything here is so positive … but here it felt like a community, and it was supportive”. The specialists agreed, emphasizing the importance of positive encouragement.

Coaches, specialists, and the researcher’s observations also reported adaptations or modifications used to help their athletes succeed. These fell into two categories: instructional strategies and task modifications. Table 2 and Table 3 document the full range of adaptations and modifications used during camp.

**Table 2 ijerph-20-05584-t002:** General Instructional Strategies.

Strategy	Data Source ^a^
Pre-teaching using tactile boards, miniature sailboat, bike parts, etc.	P, O, CB, FG
Start with physical assistance then fade assistance	P, O
Verbal description of activity (e.g., switch hand position, more power, bend your knees, reach up to the rock above your left knee, etc.)	P, O
Tactile demonstration	P, O
Physically show the participant where their peer is with their hand in biking, high ropes, kayaking, SUP, and rock climbing	P, O
Use natural areas to promote independence (e.g., using dock railing to trail down to the water)	O, CB
Describe what is happening in the environment for athletes who are blind	O
Teach SUP and kayak on land first	O, CB
SUP and kayak use physical guidance, tactile modeling, and verbal assistance on land to start	O, CB
Direct and measurable instructions (e.g., paddle 20 strokes then turn around, walk to the end of the platform then climb down the ladder, etc.)	O, CB
Physical assistance for direction and orientation	P, O, CB

^a^ Clipboards: CB; Focus Group: FG; Observations: O; Posters: P.

General Teaching Strategies

**Table 3 ijerph-20-05584-t003:** Specific Sport Modifications.

Sport	Data Source ^a^
Stand Up Paddle Board	
Kneel on knees paddling	P, O
Sitting and paddling	P, O, CB
Sitting or kneeling with coach	P, O, CB
Use a shorter paddle	P, O
Bright tape or rope on one side for shaft and blade for hand position (tactile and visual) to help orient the blade in the correct position	P, O, CB
Tape at different levels on paddle to help with depth	O, CB
Bells on leaders wrist for sound to follow	P, O, CB
Music on the canal bank boundary for direction	P, O, CB
Physical guidance and verbal assistance for strokes on land	P, O, CB
Athletes felt each part of the paddle as they were named	O, CB
Heavier board for heavier athletes and coaches	O, CB
Bright tape on parts of the board for direction	O, CB
Practice yoga on the boards	P, O, CB
Biking	
Take time to tuck in shoe laces during pre-teaching	O, CB
Single bike (after riding the tandem 2–3 laps to gather environmental information)	P, O
Mark the front and back of the helmet to help with independence	O, CB
Bright cones at major turns	P, O, CB, FG
Toe clips or Velcro to keep feet on peddle	O, CB
Verbalize when to coast	O, CB
Music/sound source to indicate stopping and starting area	O, CB
Verbally describe landmarks	O, CB
Dismount area more noticeable with bright mats and cones	O, CB
Tactile board to show areas to launch, dismount, and coast	P, O, CB
Bright vests for off-campus ride	O, CB
Braille the biking checklist	O, CB
Add sound source to bikes so riders know when a bike approaches or passes	O, CB
Teach passing etiquette (e.g., “Passing on your right”)	O, CB
Rock Climbing	
Pre-teaching using the hand-grips for athletes to feel	O, CB
Verbal cues using the clock system (e.g., “right above your knee at 12 o’clock”)	P, O, CB
Mark the front and back of the helmet to help with independence	O, CB
Feeling the rock wall and harness and practicing the descent before climbing	P, O, CB
Peer ring bell at top as part of pre-teaching the height of the wall	P, O, CB
Staff telling the climber what they are doing, and where they are (e.g., “I’m moving to your right to stay out of your way”)	O, CB
Kayaking	
Using a double or triple kayak	O, CB
Teach the correct hand positioning on the paddle with the tape marks	O, CB
Allow athletes to hear the difference in sound of the paddle entering the water correctly and incorrectly	O, CB
Sailing	
Learned how to tie a T-knot	P, O
Using miniature sailboat for pre-teaching	P, O
Steered the boat with physical assistance	P, O
Fishing	
Lighted bobbers	P, O, CB
Bobbers with sound	P, O, CB
Pre-teach with fishing pole, worms, and fake rubber fish to practice with	P, O, CB
High contrast bobber	P, O, CB
Bell at end near hook	P, O, CB
Sound buoy provides direction where to cast	P, O, CB
Bright tape indication at the edge of the water	P, O,
Fishing gloves/grip for fishing pole	P, O, CB
High Ropes Course	P, O, CB
Pre-teach with harness and verbal description of course	P, O, F
Verbal description of athletes climbing	O, CB
One person gives instruction at a time	O, CB
Hiking	
Music to guide on the trail	P, O, CB
The love of jumping worked into hiking	P, O, CB
Pole exploration and height adjustment	P, O, CB
Human guide	P, O
Cane vs. hiking poles; provide choice and alternate when to use	P, O, CB

^a^ Clipboards: CB; Focus Group: FG; Observations: O; Posters: P.

#### 2.5.1. “There Are Never Enough Words or Guidance”: Instructional Strategies

Instructional strategies include techniques used to help athletes perform successfully and independently [20]. Instructional strategies for children with visual impairments can help improve motor development and encourage socialization [21]. There are many instructional strategies that can be incorporated into physical education teaching or used by parents with their child to encourage participating in new activities or refining motor skills and activities that their child already enjoys [20,21]. A Universal Design Approach includes lesson planning prior to instruction to prepare the environment and task to ensure that a child can participate in an activity with their peers [20]. 

Verbal instructions and tactile accommodations were commonly used by both coaches and specialists as instructional strategies as noted on the posters in the dorms. Coaches and specialists agreed that communication was vital before, during, and after outdoor adventure activities. Coach Paula observed, “There are never enough words or guidance, especially with blind soccer, and just like Coach Ella said, ‘if you are not speaking the entire time, you are not speaking enough’”. Coach Maryna agreed, saying, “…we have to say what is going on and what we expect them to do. I think communication is the key and what I learned the most”. Robert’s coach added, “Being very specific. You cannot just say, ‘oh, it’s over there’”. Similarly, Steven’s coach reported that “Steven is very independent, but the verbal cues helped”. The importance of communication was emphasized not only in the focus groups, but also on the clipboards and during observations. Table 2 and Table 3 demonstrate the wide variety of instructional strategies used to help athletes succeed in outdoor adventure activities. 

Coaches and specialists also provided tactile accommodations. One specialist described how a coach mapped out the course for an athlete on their back (tactually). Michael’s coach reported that “he [Michael] gets very often distracted so he needs reminders and physical guidance”. Similarly, Dennis benefited from more tactile cues and guidance. He shared, “I had a struggle with physical movements that you should know when you were a toddler”. His coach supported him with both verbal and physical cues. “I guided him to make him try and make him more successful”.

#### 2.5.2. “Auditory Equipment Really Helped”: Task Modifications

Athletes also utilized task adaptations and modifications to successfully access outdoor adventure activities. It was noted in the researcher’s observations that there was a variety of equipment (e.g., bikes, hiking poles, miniature sailboat, and a variety of paddles in boating) that made access to each activity possible. Coaches and specialists designed these adaptations and modifications to build on athlete strengths while also addressing their needs. For athletes with some vision, coaches and specialists provided high contrast materials as noted on the clipboards. For example, specialists at stand-up paddleboarding marked the paddles with high-contrast duct tape to indicate proper hand placement and paddle alignment. The fishing specialist provided brightly colored bobbers for athletes to use, and athletes wore brightly colored vests when tandem biking around campus. 

As Elena’s coach observed, “the tactile boards were really good because she always wanted to feel first. Elena likes to be independent, and the braille instructions really helped”. Other athletes benefited from tactile representations as well. Alexander’s coach shared, “The tactile boards really helped Alexander. He is completely blind and having a physical representation of everything he was doing helped a ton”.

These accommodations were often supplemented with auditory cues and equipment. For example, the fishing specialist placed bells on the bobbers for an auditory cue, and the stand-up paddleboard specialists placed a sound source at the dock to assist athletes in navigation as noted in observations and written on the clipboards. Coaches agreed that the combination of verbal/auditory and tactile cues often benefited their athletes. Robert’s coach observed, “Robert was really into feeling out the boards and all the auditory equipment really helped”. 

### 2.6. Barriers

While most athletes discussed the benefits of outdoor adventure activities, some expressed barriers associated with accessing these activities. Subthemes of barriers included fear and anxiety, exclusion and low expectations, and lack of equipment.

#### 2.6.1. “The Hardest Part Was Getting Them to Go Out There”: Fear and Anxiety

Many athletes reported feeling fearful or anxious, especially when starting outdoor adventure activities. Julia stated, “I was very nervous to stand on the paddleboard because I did not want to fall into the water. I was even scared with my life jacket”. However, most athletes felt more confident when participating in activities. Michael’s coach shared, “Michael was a bit hesitant to do a lot of things, but when he was participating, he was really happy with each and every sport”.

The specialists agreed that some athletes expressed fear to go on their own, and according to one specialist, “the hardest part was getting them to go out there”. Some athletes were anxious about being on a bike, while others were afraid of the heights associated with the ropes course. The specialists felt, though, that “by talking them through it, the specialists and coaches were able to help the athletes to feel more comfortable...enough to not be scared anymore”. Another specialist shared that it was important to “take it slow and easy”. Coaches and specialists often faded supports as athletes became more confident and comfortable in the activity. Lieberman and colleagues [2] similarly found concern for health and safety a barrier to participation in outdoor activities. Fear and anxiety regarding participation in outdoor activities is critical to address as it could lead to avoidance of participation in outdoor activities and eventually a more sedentary lifestyle [2,7]. 

#### 2.6.2. “Make You Feel like You Are Nothing”: Exclusion and Low Expectations

Athletes are encouraged to actively and fully participate in all camp activities; however, their experiences outside of camp may differ. Elena shared, 

“Sometimes at school things can be negative and you think to yourself that you do not want to do it. They will pull you apart to make you feel like you are nothing, but here it felt like a community, and it was supportive.”

Several coaches echoed these feelings. Elena’s coach reported, 

“I talked a lot to Elena about physical activity and her gym classes and it hurts a lot because she loves physical activity but sometimes, she gets excluded from her classes mostly because she has a visual impairment and got bullied from her friends and the teachers did not help so much.” 

Similarly, Dennis’ coach shared, “With Dennis, he was just placed in Adapted PE (physical education) without being told, and I feel like a lot of people put these kids in bubbles when they can do anything”. Although physical education does not always encompass outdoor adventure these feelings are pervasive in outdoor adventure pursuits and physical education outside of camp for many of the participants.

Participating in a wide variety of outdoor activities can encourage youths to feel a part of something and increase self-determination. Rather than excluding youths with visual impairments from participating in new and challenging outdoor activities, modifications should be made when appropriate to enable the child to participate safely and with a reasonable amount of risk [6]. By encouraging youths with visual impairments to extend their boundaries in a safe manner, they are able to develop their autonomy, competence, and relatedness. Participating in activities in a safe environment will likely diminish their fear and anxiety and replace the fear with more self-determined motivation to participate in activities which they enjoy. In addition, youths with visual impairments will have more activities in which they can participate with their peers.

#### 2.6.3. “Ask for More”: Lack of Equipment

The need for specialized equipment can limit athletes’ participation in outdoor adventure activities outside of camp. Specialists noted that many of the activities that were new to athletes, such as sailing or stand-up paddleboarding, required specialized equipment that many could not access. For example, Steven shared that he had enjoyed the rock-climbing wall, so he was excited to receive a day pass from a local rock-climbing gym. 

However, camp provides a unique opportunity for athletes to experience these outdoor adventure activities. As one specialist observed, “Kids are more likely to ask for more opportunities and equipment when they go home now that they realize they can do these activities with their friends”. Several coaches agreed, and in particular, Elena’s coach stated, “She loved sailing mostly and SUP [stand-up paddleboarding] because she, well, is very good at it, she got one from her parents last year, so she has been practicing ever since”. In fact, one athlete focus group concluded with the athletes brainstorming with one another on how to acquire the equipment needed to participate in outdoor activities, such as fishing.

It should be noted here that although this was a heterogeneous camp with all athletes with a visual impairment the authors are not taking a stance on one setting over another. The fact is that this setting promoted the involvement in outdoor adventure activities with necessary accommodations. These youths now know what they need and what to ask for then they go into any school or community program.

Lack of appropriate equipment is consistent with the literature [6,22]. To assist parents, teachers, and professionals who work with individuals with visual impairments, resources and modifications have been published including modifications for outdoor adventure activities including fishing, rock climbing, kayaking, and several others [22]. Children who are exposed to a variety of outdoor recreational activities will not only feel more comfortable trying these activities at home but are much more likely to self-advocate to acquire the equipment and support they need [22,23]. A full list of sport-specific accommodations can be found in Table 3.

### 2.7. Limitations

The current study relied upon a relatively small (*n* = 37 in the Camp program, and *n* = 10 athletes in the focus groups and *n* = 12 coaches and specialists in the focus groups) convenience sample of athletes and coaches in attendance at camp, and athletes participating in the focus groups were purposively chosen. Furthermore, athletes and coaches chose to participate in camp, which focuses on sport and recreational activities, rather than academics or other ECC skills. These factors, along with the fact that data were collected over a period of one week, limit the full generalizability of the findings. 

## 3. Conclusions

Outdoor adventure activities can provide youths with visual impairments opportunities to be physically active while engaging with their peers or independently. From the present study three themes emerged: benefits, support, and barriers. While many benefits emerged from the findings, the importance of appropriate instructional strategies and task modifications was also very apparent in the findings. In addition, parents and practitioners should be aware of their own safety concerns as well as those of the youths with visual impairments as these fears may be barriers to providing and participating in outdoor adventure opportunities. It is important for all practitioners to be aware that youths with visual impairments have the right to participate in the same activities as their sighted peers and every effort should be made to provide safe opportunities for all children to participate. All children deserve the opportunity to enjoy the many benefits of outdoor adventure activities.

## Figures and Tables

**Table 1 ijerph-20-05584-t001:** Athlete Demographics.

Name	Gender	Age	Level of Visual Impairment
Robert	M	16	B3
Dennis	M	12	B1
Steven	M	13	B3
Elena	F	15	B3
Alexander	M	18	B1
Julia	F	13	B3
Andrei	M	13	B3
Inna	F	12	B4
Kira	F	17	B3
Michael	M	17	B1

Note: Level of visual impairment is based on the United States Association of Blind Athletes (n.d.) visual classifications: B1 (totally blind), B2 (20/600 and up), B3 (20/200–20/599), B4 (better than 20/199).

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
