# Peer review of "“You Feel a Sense of Accomplishment”: Outdoor Adventure Experiences of Youths with Visual Impairments during a One-Week Sports Camp"

_ijerph, 2023, doi:10.3390/ijerph20085584_

Round 1

Reviewer 1 Report

In general the paper is well written and clear, but there are some considerations: 

Focus group: It is necessary to put the inter-judge agreement of the experts in relation to the design of the group script and explain the steps for its implementation (theoretical framework on which it is based, content, etc.). 

Line 148: It is necessary to explain what the "critical friends" are and how they have been selected and why. 

Author Response

Related to the data analysis

First, the two researchers independently coded the data for themes, subthemes, and accompanying quotes to fully understand the entirety of the transcript data as well as clipboards, observation, and poster data [11]. The researchers used Braun and Clark’s (2006) recommendations related to completing thematic analyses to ensure their analysis was performed in a theoretical and systematic manner. Upon completing their initial coding, the two researchers met to discuss variations of codes, assess common codes, and review emerging themes. During these discussions the themes and subthemes were developed and continuous analysis of the data through repeated examination occurred [12].

Related to the qualification of the critical friends "who were experts in qualitative research and outdoor adventure" was added

Reviewer 2 Report

Thank you for this important paper - I have seen many about the experiences of adaptive camps for children with intellectual and developmental or physical disabilities like amputation, but this is the first I have seen about youth with visual impairments. I very much appreciate that most of the insights were drawn directly from the youth participants, rather than resorting to proxy reports from camp staff and parents/caregivers. 

As a whole, there are many instances of passive voice which can make some of the language more muddled and unclear, so I recommend revising to active voice throughout. 

The abstract in particular does not ready very polished and organized - I would especially recommend rewriting with more specifics or deleting the last sentence. 

I also noticed inconsistent reference stylings throughout, sometimes using author/date and sometimes superscripts - please refer to the preferred journal style and revise accordingly.

The problem statement of this study is unclear. I can find the purpose statement at the end of the introduction, but there needs to be a clear problem statement ahead of it to situate the purpose statement. The study could also benefit from a research question. 

In the participants section, it is unclear whether 37 was the number of children who participated, or the total number of children at the camp. Please clarify the relationship between the 37 listed, the 10 in Table 1, and the total number at camp. Also give details about any who declined to participate or were deemed ineligible. 

Replace the word "Athlete" in the Table 1 title with "Participant"

Can you please give a more clear description of the accessibility of the post-it note method for data collection? I am not sure I understand how this could be done by participants with very low vision.

In the observations section, it is noted that field notes were collected by the researchers. Please describe any training used to help the research team prepare for collecting observational data, as there can be wide variation in how this is done across individuals. 

I really like the names of the subheadings under each theme, especially benefits! Since they come from participant quotes, I would include quotation marks in the headings to make that even more apparent and give due credit to the participants for their insights.

Table 2 reads a bit messy as it has come through in the reviewer system. I think it would be easier to read if each column was left-aligned, with better spacing between rows. Same for Table 3. In Table 3 there could also be better spacing between the different sport categories. I think this Table is especially important and useful for all folks involved in adaptive outdoor recreation or camp activities.

Section 2.6.2 is missing quotation marks around each participant quote in the text. You should go through the whole results section and ensure that each quote has quotation marks, and those three full lines or longer are indented appropriately.

Author Response

As a whole, there are many instances of passive voice which can make some of the language more muddled and unclear, so I recommend revising to active voice throughout. Revised where possible

The abstract in particular does not ready very polished and organized - I would especially recommend rewriting with more specifics or deleting the last sentence. Revised

I also noticed inconsistent reference stylings throughout, sometimes using author/date and sometimes superscripts - please refer to the preferred journal style and revise accordingly.

The problem statement of this study is unclear. I can find the purpose statement at the end of the introduction, but there needs to be a clear problem statement ahead of it to situate the purpose statement. The study could also benefit from a research question. Added

In the participants section, it is unclear whether 37 was the number of children who participated, or the total number of children at the camp. Please clarify the relationship between the 37 listed, the 10 in Table 1, and the total number at camp. Also give details about any who declined to participate or were deemed ineligible. Added

Replace the word "Athlete" in the Table 1 title with "Participant" left as athlete to note these are camper/athlete participants

Can you please give a more clear description of the accessibility of the post-it note method for data collection? I am not sure I understand how this could be done by participants with very low vision. Added

In the observations section, it is noted that field notes were collected by the researchers. Please describe any training used to help the research team prepare for collecting observational data, as there can be wide variation in how this is done across individuals. We did practice observstions during the training day as a group to ensure uniformity of data collection

I really like the names of the subheadings under each theme, especially benefits! Since they come from participant quotes, I would include quotation marks in the headings to make that even more apparent and give due credit to the participants for their insights. Thank you!

Table 2 reads a bit messy as it has come through in the reviewer system. I think it would be easier to read if each column was left-aligned, with better spacing between rows. Same for Table 3. In Table 3 there could also be better spacing between the different sport categories. I think this Table is especially important and useful for all folks involved in adaptive outdoor recreation or camp activities. Revised

Section 2.6.2 is missing quotation marks around each participant quote in the text. You should go through the whole results section and ensure that each quote has quotation marks, and those three full lines or longer are indented appropriately. Revised

Reviewer 3 Report

see file attached

Author Response

  1. Why is OA effective for them?-there is no data to date in addition to what we found. This is a very new area of research in our field and we hope to do more.
  2. Addresses after line 49
  3. Focus group questions for the participants were added as Appendix A
  4. Braun & Clark (2006) added in the references
  5. Pg. 10 quotations added

6- good idea, beyond the scope of our article and will address this in a future study

7- NOT taking a stand-although this was in a self-contained setting with a more heterogeneous group, the point we are making is that these youth can take what they learned and advocate to be included in OA for their lifetime. In addition, by publishing these positive outcomes and successful adaptations we hope to promote more inclusive environments for youth with VI in the future.

Added in the article “It should be noted here that although this was a heterogeneous camp with all athletes with a visual impairment the authors are not taking a stance on one setting over another. The fact is that this setting promoted the involvement in outdoor adventure activities with necessary accommodations. These youth now know what they need and what to ask for then they go into any school or community program.”

  1. This is a blind review and this is out of line in this sense. Also, there is no rules in this journal about the number of citations by experts in the field in an article.

Round 2

Reviewer 1 Report

It´s ok

Reviewer 3 Report

Thank you for answering my queries.